# Relativistic Configuration-Interaction and Perturbation Theory Calculations of the Sn XV Emission Spectrum

Dmytro Filin, Igor Savukov * and James Colgan

Los Alamos National Laboratory, Los Alamos, NM 87545, USA; dfilin@udel.edu (D.F.); jcolgan@lanl.gov (J.C.)
* Correspondence: isavukov@lanl.gov

**Abstract:** Recently, there has been increased interest in developing advanced bright sources for lithography. Sn ions are particularly promising due to their bright emission spectrum in the required wavelength range. Cowan's code has been used to model the emission; however, it has adjustable parameters, which limit its predictive power, and it has limited relativistic treatment. Here, we present calculations based on ab initio relativistic configuration-interaction many-body perturbation theory (CI-MBPT), with relativistic corrections included at the Dirac-Fock level and core-polarization effects with the second-order MBPT. As a proof of principle that the theory is generally applicable to other Sn ions with proper development, we focused on one ion where direct comparison with experimental observations is possible. The theory can also be used for ions of other elements to predict emissions for optimization of plasma-based bright sources.

**Keywords:** CI-MBPT; configuration-interaction many-body perturbation theory; highly charged Sn ions spectra



## 1. Introduction

The resolution of lithography is limited by employed radiation sources, and currently, 193 nm light is used. High-temperature plasma containing Sn ions emitting at 13.5 nm is a promising radiation source for extreme ultra-violet lithography (see, for example, [1]). To produce such plasma, a tin droplet is arranged to fall into the interaction region where a high-power laser is focused. The absorbed laser energy is sufficient to cause a high degree of ionization of the Sn atom, over a wide range, including Sn XV (+14), considered here.

To optimize the light source and to understand the physics, it is important to model the plasma using atomic structure and kinetics codes. Initially, it was proposed that the bright emission in the 13.5 nm range occurs due to the $4p^6 4d^m - 4p^5 4d^{m+1} + 4p^6 4d^{m-1} 4f$ transition array in Sn$^{8+}$–Sn$^{14+}$[2], but it was also suggested that more excited states may contribute [3]. Most recently, Cowan's code with adjustable parameters was used, and reasonable agreement was found between theory and experiment when a very large number of dipole transitions were included (up to $10^{10}$ [4]). Opacity spectra were calculated for local thermal equilibrium (LTE) Sn plasma with a temperature of 32 eV and a density of 0.002 g/cm$^3$. The contributions were subdivided in several bands according to the energy of the lower state into which the ions radiatively decay: transitions into the ground state manifold from single-electron excited states, transitions from double-electron excited states into single-electron excited states, etc.

Preliminary ab initio relativistic configuration-interaction many-body perturbation theory (RCI-MBPT) calculations were performed, and reasonable agreement was found with the previous work for energy levels and oscillator strengths of Sn XIII-XVI [5]. The calculations were focused on the important $4p^n 4d^m - 4p^{n\pm 1} 4d^{m\mp 1} + 4p^n 4d^{m-1} 4f$ and $4p^{n-1} 4d^m 4f - 4p^{n-1} 4d^{m+1} + 4p^n 4d^{m-1} 4f$ ($n = 4$ or $5$, $m = 2$) transitions in the region of 13–14 nm that dominates the spectral emission for these tin ions. Some new lines that have significant contributions and had not been considered in previous work were

identified. A list of prominent lines was compiled in [6], where it was possible to match theoretical and experimental strong transitions after the theoretical spectrum was uniformly shifted by +0.19 eV with a residual difference in wavelength on the order of 0.02 nm. However, the theoretical spectrum had significant deviations from experiment especially in the longer-wavelength region above 13.7 nm.

The RCI-MBPT method is well suited for these highly ionized Sn ions because perturbation theory becomes more accurate with the ion charge $Z$, and relativistic effects are included at least at the Dirac–Hartree–Fock level, with the dominant part of Breit interaction between valence and core electrons included. For example, a good accuracy has been achieved for the forbidden electric-dipole transitions of beryllium-like ions, where relativistic effects are important [7]. In addition, neon-like ions, to some extent similar to the present case of the $Sn^{14+}$ ion, were also investigated using the particle-hole CI-MBPT, and consistent agreement was found when the Breit contribution was included in the self-consistent field calculation [8,9]. Unfortunately, this specific particle-hole theory was based on CI of single-excitation particle-hole states, but the emission spectrum of the $Sn^{14+}$ ion requires consideration of states with double excitations. Therefore, instead, a more general CI-MBPT theory that can include configurations with single, double, etc. excitations developed by Dzuba et al. (see [10]) is used in the current work.

## 2. Theory

### 2.1. CI-MBPT Framework

The CI-MBPT method relies on the subdivision of atomic electrons into valence and core electrons, with valence electrons having a much smaller binding energy than core electrons. The interaction between core and valence electrons is much weaker than between valence electrons. the CI-MBPT method accurately treats valence–valence interactions via CI and includes important valence–core interactions in the second order of MBPT, significantly saving numerical cost compared with the inclusion of these interactions in CI.

To calculate energies and wavefunctions, a CI-MBPT method developed for open shell atoms with multiple valence electrons (see, e.g., [10]) is used. The effective CI-MBPT Hamiltonian for an atom is split into two parts:

$$H^{eff} = \sum_{i=1}^{M} h_{1i} + \sum_{i \neq j}^{M} h_{2ij}.$$

(1)

The one-electron contribution is

$$h_1 = c\boldsymbol{\alpha} \cdot \mathbf{p} + (\beta - 1)mc^2 - Ze^2/r + V^{N-M} + \Sigma_1.$$

(2)

Here, the $4 \times 4$ Dirac matrices $\boldsymbol{\alpha} = \begin{pmatrix} 0 & \sigma \\ \sigma & 0 \end{pmatrix}$ and $\beta = \begin{pmatrix} 0 & 1 \\ 1 & 0 \end{pmatrix}$, with "1" and "0" being the unit and zero $2 \times 2$ matrices, and $\sigma$ is the Pauli matrices. In addition to the $V^{N-M}$ ($N$ and $M$ are the numbers of all and valence electrons, respectively) DHF part, the potential contains the valence electron self-energy correction, $\Sigma_1$ [11]. In the CI-MBPT program, the self-energy correction is calculated with the second-order MBPT. The two-electron Hamiltonian is

$$h_2 = e^2/|\mathbf{r_1} - \mathbf{r_2}| + \Sigma_2$$

(3)

where $\Sigma_2$ is the Coulomb interaction screening term arising from the presence of the core [12], which is calculated in the second order of MBPT. Details on the CI+MBPT approach can be found in Reference [13].

### 2.2. CI-MBPT Numerical Procedure

In numerical calculations, first, the DHF $V^{N-2}$ starting potential is calculated and contains the contributions from $4p^4$ electrons. The maximum radius in our DHF calculations (we use the spherical coordinate system centered on the nucleus) was chosen to be 15 a.u.,

which is sufficient to cover the extent of wavefunctions needed for transition calculations. The Breit potential was also included in the DHF calculations, resulting in the so-called Breit DHF wavefunctions. The spline subroutine has a maximum angular momentum of basis orbital 5 and the cavity radius 15 a.u. All of the core states (the states below 4s) were included into MBPT calculations. Random-phase approximation was included. The maximum angular momentum of the Coulomb integrals to be included into CI was 4.

The final step in the calculations of energy states and wavefunctions is the the solution of the eigenvalue problem for the effective Hamiltonian matrix that includes the second-order MBPT corrections.

In the ab initio CI-MBPT, the scaling MBPT coefficients (the coefficients in front of $\Sigma_1$ and $\Sigma_2$ terms that can be introduced to improve agreement with the experiment for energies) are set to one. It works well for light atoms, in which valence–core interactions are sufficiently small to be taken into account in the second order of MBPT. For example, Si I ab initio CI-MBPT calculations resulted in close agreement with the experiment for energies and transition probabilities [14,15]. Good agreement is also found in CI-MBPT calculations for Ge, Sn, and Pb [10], although the level of accuracy is decreased.

### 2.3. Input Configurations

The non-relativistic configurations used in the input of CI-MBPT program are listed for in Table 1 for even parity and in Table 2 for odd parity. The even configurations were obtained by single, double, and triple excitations of the ground state configuration $4s^2 4p^6$. Single excitations were from the 4s to 5s, 6s, 4d, 5d or 4p to 4f–6f, 5p, 6p states. Double excitations were various combinations of excitations of 4s and 4p electrons to the first and second unoccupied shells. Some triple and even quadruple excitations were also included. The inclusion criterion was the size of the contribution. The odd configurations were generated by promoting 4s to 4f, 5f, 5p, 6p or 4p to 4d, 5d, 5s, 6s for single excitations, and various combinations of 4s and 4p promotions for double excitations. As in the case of the even states, some triple and quadruple excitations were included on the basis of their contribution. The automatic excitation algorithm was not used because the number of determinants would become too large for computation. The list of non-relativistic configuruations was converted into the list of relativistic configurations automatically. There was no attempt to downsize the list of resulting relativistic configurations.

**Table 1.** Input configurations for even states.

| | | | | | | | |
|---|---|---|---|---|---|---|---|
| 1 | $4s^2 4p^6$ | 12 | $4s4p^5 4d4f$ | 23 | $4s^2 4p^4 4f^2$ | 34 | $4s4p^4 4d5s^2$ |
| 2 | $4s4p^6 5s$ | 13 | $4s4p^5 4d5f$ | 24 | $4s^2 4p^4 4d^2$ | 35 | $4s4p^4 4d5p^2$ |
| 3 | $4s4p^6 6s$ | 14 | $4s4p^5 4d5p$ | 25 | $4s^2 4p^4 4d5d$ | 36 | $4s4p^4 4d5d^2$ |
| 4 | $4s4p^6 4d$ | 15 | $4s4p^5 4d6p$ | 26 | $4s^2 4p^4 5p4f$ | 37 | $4s4p^4 4d5p4f$ |
| 5 | $4s4p^6 5d$ | 16 | $4s4p^5 5d5p$ | 27 | $4s^2 4p^4 5p6p$ | 38 | $4s^2 4p^3 4d^2 4f$ |
| 6 | $4s4p^6 6d$ | 17 | $4s4p^5 5d6p$ | 28 | $4s^2 4p^4 4d5s$ | 39 | $4s^2 4p^3 4d^2 5p$ |
| 7 | $4s^2 4p^5 4f$ | 18 | $4s4p^5 5s5p$ | 29 | $4s^2 4p^4 5d5s$ | 40 | $4s^2 4p^3 4d5s4f$ |
| 8 | $4s^2 4p^5 5f$ | 19 | $4s4p^5 5s6p$ | 30 | $4s^2 4p^4 4d6s$ | 41 | $4s^2 4p^3 4d5s5p$ |
| 9 | $4s^2 4p^5 6f$ | 20 | $4s4p^5 6s5p$ | 31 | $4s4p^4 4d^2 5d$ | 42 | $4s4p^3 4d^3 4f$ |
| 10 | $4s^2 4p^5 5p$ | 21 | $4s4p^5 5s4f$ | 32 | $4s4p^4 4d^3$ | 43 | $4s^2 4p^2 4d^4$ |
| 11 | $4s^2 4p^5 6p$ | 22 | $4s^2 4p^4 5p^2$ | 33 | $4s4p^4 4d4f^2$ | 44 | $4p^6 4d^2$ |

**Table 2.** Input configurations for odd states.

| 1 | $4s4p^64f$ | 13 | $4s4p^54f^2$ | 25 | $4s^24p^45s4f$ | 37 | $4s^24p^34d4f^2$ |
|---|---|---|---|---|---|---|---|
| 2 | $4s4p^65f$ | 14 | $4s4p^55p^2$ | 26 | $4s^24p^45s5f$ | 38 | $4s^24p^34d^3$ |
| 3 | $4s4p^65p$ | 15 | $4s\,4p^54d5s$ | 27 | $4s^24p^46s4f$ | 39 | $4s^24p^34d^25d$ |
| 4 | $4s4p^66p$ | 16 | $4s\,4p^54d5d$ | 28 | $4s^24p^46s5f$ | 40 | $4s^24p^34d^25s$ |
| 5 | $4s^24p^54d$ | 17 | $4s\,4p^55p4f$ | 29 | $4s^24p^44d4f$ | 41 | $4s^24p^34d^26s$ |
| 6 | $4s^24p^55s$ | 18 | $4p^54d^3$ | 30 | $4s^24p^44d5f$ | 42 | $4s^24p^34d5p4f$ |
| 7 | $4s^24p^56s$ | 19 | $4s^24p^44d5p$ | 31 | $4s4p^44d^24f$ | 43 | $4s4p^34d^4$ |
| 8 | $4s^24p^55d$ | 20 | $4s^24p^44d6p$ | 32 | $4s4p^44d^25f$ | 44 | $4s4p^34d^24f^2$ |
| 9 | $4s^24p^56d$ | 21 | $4s^24p^45d5p$ | 33 | $4s4p^44d^25p$ | 45 | $4s4p^34d^25s^2$ |
| 10 | $4s4p^55s^2$ | 22 | $4s^24p^45d6p$ | 34 | $4s4p^44d^26p$ | 46 | $4s4p\,^34d^25p^2$ |
| 11 | $4s4p^56s^2$ | 23 | $4s^24p^45s5p$ | 35 | $4s4p^44d5s4f$ | 47 | $4s4p^34d^25d^2$ |
| 12 | $4s4p^54d^2$ | 24 | $4s^24p^45s6p$ | 36 | $4s^24p^34d5p^2$ | | |

### 2.4. Wavelengths and gf-Values

Wavelengths and $gf$-values of prominent transitions were previously reported in [6]. Here, we provide a comparison with our CI-MBPT values in Table 3. The CI-MBPT code outputs relativistic Landé $g$-factors, and we used them together with the non-relativistic values obtained from $S$, $L$, and $J$ values of the non-relativistic $LS$-coupling terms based on the expression:

$$g_{nr} = 1 + \frac{J(J+1) - L(L+1) + S(S+1)}{2J(J+1)} \tag{4}$$

to assign terms to the relativistic states. In most cases, we obtained agreement with the terms assigned using Cowan's code, for example given by [6], but some transitions were spin-forbidden if the terms of [6] were used, so we replaced some terms involved with those that would make strong transitions spin-conserving. In addition, some transitions were not listed by [6] and we used this principle to determine the missing term notations. Due to significant relativistic effects leading to singlet-triplet mixing, resulting in a deviation of relativistic $g$-factors from non-relativistic ones, there is some ambiguity in the $LS$-coupling term assignment.

In general, there is some agreement for $gf$-values and a better than one percent agreement for wavelengths. As CI-MBPT includes relativistic effects more consistently and core-polarization effects, we believe that our calculations of $gf$-values are more accurate. The wavelength inaccuracy of our method can be attributed to neglected higher-order MBPT corrections, beyond the included second order. Additionally, many closely spaced levels can cause some variation in the the predicted wavelength due to configuration interactions. The perturbation theory converges as $1/Z$, and with $Z = 14$, we expect the energy accuracy of $1/Z^2 = 0.005$. It is also important to note that there is some instrumental inaccuracy of 0.01 nm of the experimental wavelengths [6]. Figure 1 shows the ratio of [6] and CI-MBPT gf values, which indicates that a systematic value of about 1.3 and small scatter is observed for large $gf$ values. The scatter increases for smaller $gf$ values, as expected due to lower accuracy. The accuracy of CI-MBPT for strong transitions can be estimated at about 10%, although Kr-like, Ar-like, and Ne-like ions are quite sensitive to relativistic and correlation effects. For example, relativistic particle-hole CI-MBPT [9] achieved about 10% agreement with the experiment in the case of Ne-like ions, but there is significant experimental uncertainty. Some estimate of accuracy of the CI-MBPT $gA$ and related $gf$ values can be obtained from the comparison of CI-MBPT and experimental emission intensities (the next section). It is also interesting to note that our CI-MBPT $gA$ values of $1.64 \times 10^{12}$ s$^{-1}$ for $^1S_0 - ^1P_1$ transition agrees quite well with the $gA$ value of [16] $1.8 \times 10^{12}$ s$^{-1}$, much better the comparison of our $gf$ value with that of [6], 4.3 vs. 5.8. It is

interesting to note that, in [17], it was pointed out that the wavelength of the resonant transition $^1S_0 - ^1P_1$ was incorrectly reported to be 13.2643 nm [18], while the value reported in [17] was 13.3431 nm, in good agreement with [19], where the isoelectronic sequence of Kr was systematically studied. Our calculations agree better with the 13.2643 nm wavelength, but this may be due to limited accuracy of our ab initio calculations of energies (especially a relatively large systematic shift between the odd and even levels), while Cowan's calculations have several adjustable parameters to improve the accuracy of energy levels. Transition amplitudes, on the other hand, are expected to be more accurate in the CI-MBPT calculations.

**Table 3.** Comparison of our CI-MBPT calculations with results of [6] (calculations with Cowan's code and experimental measurements) for wavelengths ($\lambda$ in nm) and $gf$ values of Sn XV prominent $4s^24p^54d - 4s^24p^44d^2 + 4s^24p^54f$ transitions. The CI-MBPT relativistic ($g_{rl}$) and term-based non-relativistic ($g_{nr}$) g-factor values are also listed. The term labels in the table are in some cases different from the ones of [6]. $S_{th}$ is the theoretical line strength in atomic units.

| Lower State Config. | CI-MBPT $g_{rl}$ | $g_{nr}$ | Upper State Config. | CI-MBPT $g_{rl}$ | $g_{nr}$ | $S_{th}$ | CI-MBPT $gf_{th}$ | $\lambda_{th}$ | [6] $\lambda_{th}$ | [6] $\lambda_{exp}$ | [6] $gf_{th}$ |
|---|---|---|---|---|---|---|---|---|---|---|---|
| $4s^24p^6\,^1S_0$ | 0 | 0 | $4s^24p^54d\,^1P_1^o$ | 0.979 | 1.000 | 1.87 | 4.3 | 13.23 | 13.25 | 13.29 | 5.8 |
| $4s^24p^54d\,^3D_3^o$ | 1.215 | 1.333 | $4s^24p^44d^2\,^3F_4$ | 1.123 | 1.25 | 4.64 | 10.6 | 13.29 | 13.35 | | 14.65 |
| $4s^24p^54d^3F_4^o$ | 1.250 | 1.250 | $4s^24p^54f^3G_5$ | 1.192 | 1.200 | 7.21 | 16.45 | 13.31 | 13.34 | 13.34 | 21.22 |
| $4s^24p^54d\,^3F_4^o$ | 1.250 | 1.250 | $4s^24p^44d^2\,^3F_4$ | 1.199 | 1.250 | 2.47 | 5.61 | 13.38 | 13.46 | | 11.16 |
| $4s^24p^54d\,^1D_2^o$ | 0.950 | 1.000 | $4s^24p^44d^2\,^1F_3$ | 1.051 | 1.083 | 3.6 | 8.18 | 13.39 | 13.44 | | 7.54 |
| $4s^24p^54d\,^1F_3^o$ | 1.112 | 1.000 | $4s^24p^54f\,^1G_4$ | 1.076 | 1.000 | 5.68 | 12.86 | 13.4 | 13.49 | | 16.54 |
| $4s^24p^54d\,^3D_2^o$ | 1.197 | 1.167 | $4s^24p^44d^2\,^3F_3$ | 1.069 | 1.083 | 2.14 | 4.85 | 13.41 | 13.42 | 13.46 | 5.5 |
| $4s^24p^54d\,^3F_3^o$ | 1.090 | 1.083 | $4s^24p^54f\,^3G_4$ | 1.075 | 1.050 | 4.15 | 9.39 | 13.42 | 13.54 | | 15.43 |
| $4s^24p^54d\,^3P_1^o$ | 1.417 | 1.500 | $4s^24p^44d^2\,^3D_2$ | 1.085 | 1.167 | 1 | 2.26 | 13.44 | | | |
| $4s^24p^54d\,^3P_2^o$ | 1.369 | 1.500 | $4s^24p^44d^2\,^3P_2$ | 1.378 | 1.500 | 2.35 | 5.31 | 13.45 | 13.45 | | 7.08 |
| $4s^24p^54d\,^3P_2^o$ | 1.369 | 1.500 | $4s^24p^54f\,^3F_3$ | 1.120 | 1.083 | 2.32 | 5.23 | 13.47 | 13.63 | | 4.67 |
| $4s^24p^54d\,^1D_2^o$ | 0.950 | 1.000 | $4s^24p^44d^2\,^1D_2$ | 0.890 | 1.000 | 2.3 | 5.17 | 13.49 | 13.55 | 13.57 | 6.44 |
| $4s^24p^54d\,^3F_2^o$ | 0.817 | 0.667 | $4s^24p^44d^2\,^3F_2$ | 0.794 | 0.667 | 2.27 | 5.11 | 13.5 | 13.54 | | 7.07 |
| $4s^24p^54d\,^3D_2^o$ | 1.197 | 1.167 | $4s^24p^44d^2\,^3D_2$ | 1.115 | 1.167 | 2.45 | 5.51 | 13.52 | 13.53 | | 7.74 |
| $4s^24p^54d\,^3D_3^o$ | 1.215 | 1.333 | $4s^24p^44d^2\,^3F_3$ | 1.196 | 1.083 | 4 | 8.99 | 13.52 | 13.47 | | 11.88 |
| $4s^24p^54d\,^3F_2^o$ | 0.817 | 0.667 | $4s^24p^54f\,^3G_3$ | 0.977 | 0.750 | 2.18 | 4.89 | 13.53 | 13.68 | | 10.66 |
| $4s^24p^54d\,^3D_1^o$ | 0.604 | 0.500 | $4s^24p^44d^2\,^3D_1$ | 0.770 | 0.500 | 1.81 | 4.05 | 13.55 | 13.53 | 13.53 | 5.25 |
| $4s^24p^54d\,^1F_3^o$ | 1.112 | 1.000 | $4s^24p^44d^2\,^1F_3$ | 1.069 | 1.000 | 2.52 | 5.64 | 13.57 | 13.54 | | 8.68 |
| $4s^24p^54d\,^3D_2^o$ | 1.197 | 1.167 | $4s^24p^44d^2\,^3P_1$ | 1.279 | 1.500 | 1.57 | 3.51 | 13.59 | 13.62 | 13.62 | 4.58 |
| $4s24p^54d\,^3F_2^o$ | 0.817 | 0.667 | $4s^24p^54f\,^3F_3$ | 1.005 | 1.083 | 2.31 | 5.15 | 13.61 | | | |
| $4s^24p^54d\,^3D_3^o$ | 1.215 | 1.333 | $4s^24p^44d^2\,^3F_4$ | 1.090 | 1.250 | 1.18 | 2.63 | 13.62 | | | |
| $4s24p^54d\,^3P_1^o$ | 1.417 | 1.500 | $4s^24p^54f\,^3D_2$ | 1.108 | 1.167 | 1.16 | 2.6 | 13.62 | | | |
| $4s^24p^54d\,^1P_1^o$ | 0.979 | 1.000 | $4s^24p^44d^2\,^1D_2$ | 0.996 | 1.000 | 6.03 | 13.43 | 13.63 | 13.66 | 13.65 | 18.14 |
| $4s^24p^54d\,^3D_3^o$ | 1.25 | 1.333 | $4s^24p^44d^2\,^3F_4$ | 1.170 | 1.250 | 1.85 | 4.11 | 13.65 | | | |
| $4s^24p^54d\,^3D_3^o$ | 1.215 | 1.333 | $4s^24p^44d^2\,^3D_2$ | 1.086 | 1.167 | 1.54 | 3.42 | 13.68 | 13.75 | 13.76 | 3.18 |
| $4s^24p^54d\,^3D_1^o$ | 0.604 | 0.500 | $4s^24p^54f\,^3F_2$ | 1.178 | 0.667 | 0.5 | 1.11 | 13.7 | 13.82 | 13.79 | 4 |
| $4s^24p^54d\,^3D_2^o$ | 1.197 | 1.167 | $4s^24p^54f\,^3F_3$ | 0.977 | 1.083 | 1.2 | 2.64 | 13.85 | | | |
| $4s^24p^54d\,^3D_2^o$ | 1.197 | 1.167 | $4s^24p^54f\,^3F_3$ | 1.005 | 1.083 | 0.91 | 1.99 | 13.94 | 14.05 | 14.03 | 5.44 |
| $4s^24p^54d\,^3F_3^o$ | 1.112 | 1.083 | $4s^24p^54f\,^3F_3$ | 1.005 | 1.083 | 1.02 | 2.19 | 14.11 | 14.2 | 14.17 | 3.71 |
| $4s^24p^54d\,^1D_2^o$ | 0.817 | 1.000 | $4s^24p^44d^2\,^1P_1$ | 1.030 | 1.000 | 1.34 | 2.87 | 14.22 | 14.27 | 14.26 | 3.45 |
| $4s^24p^54d\,^1F_3^o$ | 1.112 | 1.000 | $4s^24p^44d^2\,^1D_2$ | 1.005 | 1.000 | 1.21 | 2.58 | 14.32 | 14.39 | 14.36 | 4.98 |

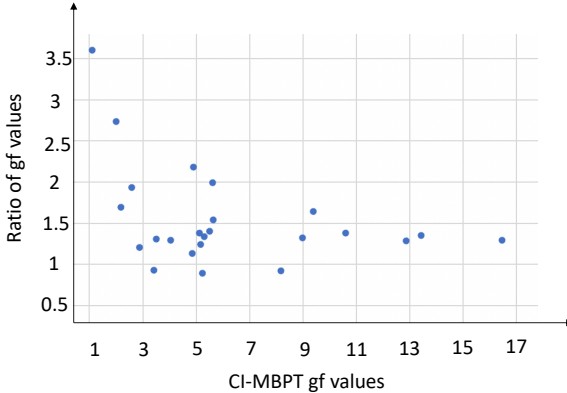

**Figure 1.** The ratio of Sn XV $gf$ values from [6] and from ab initio CI-MBPT. A systematic value of about 1.3 and small scatter can be observed for large $gf$ values. The scatter increases for smaller $gf$ values.

### 2.5. CI-MBPT Intensities

In Figure 2, we show a comparison of the CI-MBPT results with the experimental observations [6]. The CI-MBPT $gA$ values, which are proportional to the intensities of transitions, were multiplied by a single constant to have a closely matching fit, and the experimental intensities were normalized to the maximum. In the calculations, the maximum number of states, $n_{max}$ for each $J$ was limited to 40. The considered $J$ were from 0 to 13 for odd and 0 to 12 for even states. Multiple bands were found for all of the considered odd and even states limited by $n = 40$. To build the spectrum, 993 lines were used totally. A small shift of 0.423 eV between the odd and even states was introduced to bring the theoretical results in better agreement with the experimental observations. As the intensities are proportional to $gA$ values, we calculated $gA$ values and used a single scaling factor for best fit. Local temperature equilibrium (LTE) was not used with a temperature as a fitting parameter because cascade processes could lead to significant deviations from LTE and excessive population of $4s^24p^44d^2$, $4s4p^44d^3$, $4s4p^54d4f$, $4s^24p^34d^24f$ even states and $4s^24p^54d$, $4s4p^54d^2$, $4s^24p^34d^3$, $4s^24p^44d4f$, $4s4p^44d^24f$ odd states, transition from which are very important for the considered spectrum range. We included the array of these states and nearby states within 40 eigen-energies of each $J$. It is worth noting that transitions between states with high $J$, such as $J = 5, 6, 7, 8$ and 9, play a prominent role in the spectrum. The broadening of $\sigma = 0.025$ nm (the line profile is $f(x) = \frac{1}{\sigma\sqrt{2\pi}}exp[-\frac{(x-\lambda)^2}{2\sigma^2}]$) was used to account for the overall instrumental resolution of about 0.01 nm [6] and other possible broadening mechanisms.

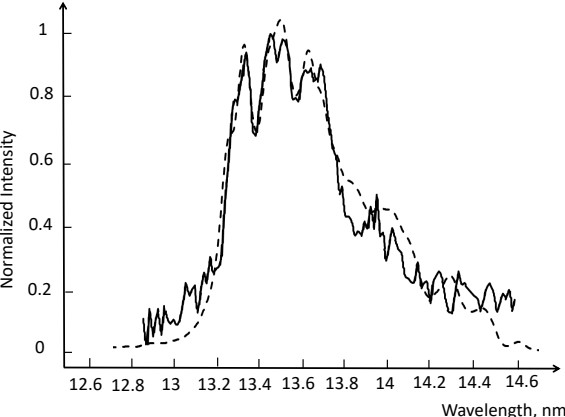

**Figure 2.** Comparison of observed (solid line) of [6] and the present ab initio CI-MBPT (dash) spectra of emissions of Sn XV. The intensity is normalized to the maximum intensity of the observed spectrum. The CI-MBPT spectrum was obtained by calculating $gA$ values between odd and even level bands, with some shift of the first odd band resulting in a 0.06 nm wavelength shift.

According to D'Arcy et al. [6], the full experimental spectrum is dominated by an intense emission array that contains the strongest lines of Sn VII–XV identified by Churilov and Ryabtsev [20,21].

### 3. Conclusions

We calculated the emission intensity profile of $Sn^{14+}$ ions and compared the results with previous experimental observations. A close agreement was found, which indicates that the relativistic CI-MBPT method is suitable for the analysis of emission from a plasma containing $Sn^{14+}$ ions. The theory can be applied to other similar ions to understand the full emission spectrum of plasmas that have potential uses in lithography.

**Author Contributions:** D.F. performed CI-MBPT calculations, plotted the graph for comparison with the experiment, and compiled the table. I.S. was involved in contributing with text and discussions of the results, and in communication to the journal, as the corresponding author. J.C. was guiding this research, was involved in discussions and proof reading. All authors have read and agreed to the published version of the manuscript.

**Funding:** Research presented in this article was supported by the Laboratory Directed Research and Development program of Los Alamos National Laboratory under the project number 20180125ER.

**Institutional Review Board Statement:** Not applicable.

**Informed Consent Statement:** Not applicable.

**Data Availability Statement:** Data will be provided upon request.

**Acknowledgments:** The authors are grateful to V. A. Dzuba of the University of New South Wales, Sydney, Australia, for making the CI-MBPT code available for this work.

**Conflicts of Interest:** The authors declare no conflict of interest.

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
