# Peer review of "Relativistic Configuration-Interaction and Perturbation Theory Calculations of the Sn XV Emission Spectrum"

_atoms, doi:10.3390/atoms9040096_

Round 1

Reviewer 1 Report

Comments have been written in the file attached mm 1 Paper of atoms.....

Author Response

Dear Reviewer,

thank you for your effort and constructive critique,

please see the attached file for the responses to your comments

Authors

Reviewer 2 Report

The authors of this article, claim the emission spectrum of Sn XV is important, that the best method for analyzing the intensity of the emission spectrum is the  CI_MBPT program, and by making certain assumptions using this code,  the computed gA values for transitions between odd and even bands,  agree well with  the  normalized observed intensity as shown in their figure, the only result reported in the article. All this may well be true. Thus this research  is a step in the procedure for understanding the plasma processes  taking place in the experiment.

The paper should include the computed energy levels  (before results are adjusted for one reason or another)  as well as the most important A-values.  Some computed wave lengths were reported in Ref. 4.  Were the present results more accurate?  One would hope so.   In other words, sufficient data should be provided so that future developments can evaluate their relevant accuracy.

Some specific comment are given below.

  1. Line 81 “resulted in close agreement with experimemt for energies and transition probabilities”    What is mean by “close agreement”?

  2. Line 91 “size of contribution”  What is meant  -- the expansion coefficient , the contribution to the energy/ By how much?

  3. Line 98 “number of deterrminants became too large.. “  What is the meaning of ‘too large”

  4. Line 109 : ‘for best fist”  should be “for best fit”.

Etc.

As for the figure, I would have thought the computed intensity would be more irreguarl as well. 

Author Response

Dear Reviewer, 

thank you very much for your evaluation of our manuscript. Please find the responses to comments in the attached file,

Authors

Reviewer 3 Report

This is not science but mockery of it. Publication of such content would poorly reflect on scientific reputation not only of the authors, but also of the journal and of entire atomic physics. This very poorly presented study does not have any results except for an engineered figure comparing some suitably adjusted fragments of the calculation output (for Sn XV only) with an experimental spectrum copied from Fig. 1 of Ref. [4], which contains emission from several ionization stages of Sn. That spectrum was obtained in charge-exchange collisions with He target atoms using an ECR source. It is very different from laser-produced plasmas used for EUV lithography, which is the main motivation for the present work. The conclusions are not corroborated by any numerical data from the calculation, and the authors present no estimates of uncertainties of those numerical results.

In addition, there are numerous lapses in language and style. Some of them are outlined in the accompanying pdf file (visible in Adobe PDF Reader). My recommendation is: reject this manuscript from publication in Atoms and notify all co-authors, not only the corresponding author, about the present review. I sincerely doubt that I. Savukov and J. Colgan ever saw this manuscript.

Author Response

Dear Reviewer,

thank you very much for detailed corrections of the manuscript! We made all corrections you suggested. We believe that manuscript became much better. Please see the attached response pdf file.

As for your comments that the co-authors did not read the manuscript, everybody on the author list read and approved it.

Thank you again, we hope that now you would approve our manuscript as written more rigorously,

The authors

Round 2

Reviewer 2 Report

Line 34:  Omit the 4s^2

Line  61: Where is the exchange between the outer n+m outer electrons with the core included

Table 3.  Caption should explain the meaning of lambda

Reviewer 3 Report

The manuscript has notably improved in the last revision. However, more revisions are still required before it can be accepted for publication. See the detailed review in the accompanying Word and pdf files. 
